# Social fluidity mobilizes contagion in human and animal populations

**Ewan Colman[1,2]\*, Vittoria Colizza[3], Ephraim M Hanks[4], David P Hughes[5], Shweta Bansal[1]\***

[1]Department of Biology, Georgetown University, Washington, United States; [2]Roslin Institute, University of Edinburgh, Midlothian, United Kingdom; [3]INSERM, Sorbonne Université, Institut Pierre Louis d'Épidémiologie et de Santé Publique (IPLESP UMRS 1136), F75012, Paris, France; [4]Department of Statistics, Eberly College of Science, Penn State University, State College, United States; [5]Department of Entomology, College of Agricultural Sciences, Penn State University, State College, United States

**\*For correspondence:**
ecolman@ed.ac.uk (EC);
shweta.bansal@georgetown.edu (SB)

**Competing interests:** The authors declare that no competing interests exist.

**Abstract** Humans and other group-living animals tend to distribute their social effort disproportionately. Individuals predominantly interact with a small number of close companions while maintaining weaker social bonds with less familiar group members. By incorporating this behavior into a mathematical model, we find that a single parameter, which we refer to as *social fluidity*, controls the rate of social mixing within the group. Large values of social fluidity correspond to gregarious behavior, whereas small values signify the existence of persistent bonds between individuals. We compare the social fluidity of 13 species by applying the model to empirical human and animal social interaction data. To investigate how social behavior influences the likelihood of an epidemic outbreak, we derive an analytical expression of the relationship between social fluidity and the basic reproductive number of an infectious disease. For species that form more stable social bonds, the model describes frequency-dependent transmission that is sensitive to changes in social fluidity. As social fluidity increases, animal-disease systems become increasingly density-dependent. Finally, we demonstrate that social fluidity is a stronger predictor of disease outcomes than both group size and connectivity, and it provides an integrated framework for both density-dependent and frequency-dependent transmission.

## Introduction

Social behavior is fundamental to the survival of many species. It allows the formation of social groups providing fitness advantages from greater access to resources and better protection from predators (*Krause and Ruxton, 2002*). Structure within these groups can be found in the way individuals communicate across space, cooperate in sexual or parental behavior, or clash in territorial or mating conflicts (*Hinde, 1976*). While animal societies are usually studied independently of each other, studying how they differ in these regards has potential to reveal new insights into the nature of social living (*Sah et al., 2018*; *Dunbar and Shultz, 2010*).

When social interaction requires shared physical space it can also be a conduit for the transmission of infectious disease (*Altizer et al., 2003*). In a typical infectious disease model, if the disease spreads through the environment then the transmission rate is assumed to scale proportionally to the local population density (*de Jong et al., 1995*; *Hopkins et al., 2020*). Alternatively, if transmission requires close proximity encounters that only occur between bonded individuals then we expect social connectivity to determine the outcome. These two paradigms are known in the literature as density-dependence and frequency-dependence (*Silk et al., 2017*).

The problem, however, is that real diseases are not so easy to categorize (*Patterson and Ruckstuhl, 2013*). For example, as social groups grow in size, new bonds must be created to maintain cohesiveness (*Lehmann et al., 2007*). To manage the time and cognitive effort required to create these bonds, individuals tend to interact mostly with a small number of close companions while maintaining cohesion with the wider group through less frequent contact (*Silk, 2007*; *Sueur et al., 2011*; *Dakin and Ryder, 2020*). For an infectious disease, this creates fewer transmission opportunities than we would expect to see in a group with highly fluid social dynamics. The extent to which group size amplifies the transmission rate therefore depends on how individuals choose to distribute their social effort between strong and weak ties (*Karsai et al., 2014*).

While transmission rate has been observed to scale non-linearly with group size for a number of disease systems (*Cross et al., 2013*; *Smith et al., 2009*; *Silk et al., 2017*), it remains unclear how much this dependency is related to the internal social structure of the group; few studies observe social dynamics at sufficient detail while simultaneously monitoring the disease status of each individual. In the absence of direct observations, our contribution to this discussion centers around modeling; incorporating empirical social data with computational simulations. We address two specific questions. Firstly, can we quantify the variability in how individuals choose to distribute their social effort within a group, and secondly, what will this tell us about the effect that population density has on disease transmission?

In the first part of this paper, we introduce a mathematical model founded on the concept of *social fluidity* which we define as variability in the amount of social effort the individual invests in each member of their social group. Using openly available data, we estimate the social fluidity of 57 human and animal social systems. In the second part, we derive an expression for the basic reproductive number of an infectious disease in the social fluidity model and demonstrate its accuracy in predicting simulated outcomes. Furthermore, social fluidity emerges as a coherent mathematical framework providing the smooth connection between density-dependent and frequency-dependent disease systems.

## Characterizing social behavior

Our first objective is to measure social behavior in a range of human and animal populations. We start by introducing a model that captures a hidden element of social dynamics: how individual group members distribute their social effort. We mathematically describe the relationships between social variables that are routinely found in studies of animal behavior, the number of social ties and the number of interactions observed, and apply the model to empirical data to reveal behavioral differences between several species.

## Social behavior model

Consider a closed system of $N$ individuals and a set of interactions between pairs of individuals that were recorded during some observation period. These observations can be represented as a network: each individual, $i$, is a *node*; an *edge* exists between two nodes $i$ and $j$ if at least one interaction was observed between them; the *edge weight*, $w_{i,j}$, denotes the number of times this interaction was observed. The total number of interactions of $i$ is denoted *strength*, $s_i = \sum_j w_{i,j}$, and the number of nodes with whom $i$ is observed interacting is its *degree*, $k_i$ (*Barrat et al., 2004*).

We define $x_{j|i}$ to be the probability that an interaction involving $i$ will also involve node $j$. Therefore, the probability that at least one of these interactions is with $j$ is $1 - (1 - x_{j|i})^{s_i}$. The main assumption of the model is that the values of $x_{j|i}$ over all $i,j$ pairs are distributed according to a probability distribution, $\rho(x)$. Thus, if a node interacts $s$ times, the marginal probability that an edge exists between that node and any other given node in the network is

$$\Psi(s) = 1 - \int \rho(x)(1 - x)^s dx. \tag{1}$$

Technically, $\rho(x)$ is the distribution of marginal $x_{i|j}$ values of the joint probability distribution $\rho(x)$ where $X$ is a matrix whose $i,j$ entry is -1 if $i = j$ and $x_{j|i}$ otherwise. While the values of $x_{j|i}$ are subject to network interdependencies, specifically $AX = X^T A$ and $\mathbf{1} = \mathbf{0}$, where $A$ is any diagonal matrix with positive entries, and $\mathbf{1}$ and $\mathbf{0}$ are column vectors of length $N$ containing only 0 and 1, we do not take these constraints into account when estimating $\rho$.

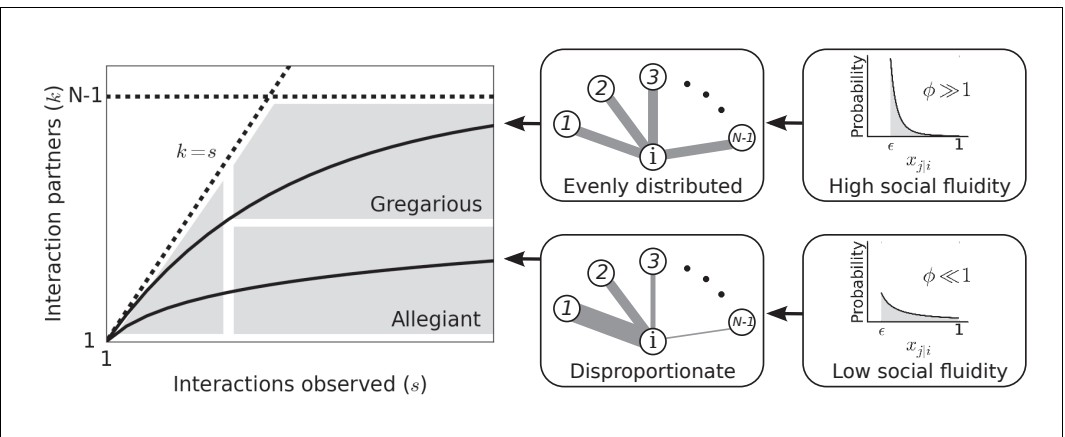

**Figure 1.** Left: Each individual can be represented as a single point on this plot. Dashed lines mark the boundary of the region where data points can feasibly be found. The mean degree is plotted for two values of $\phi$ representing two possible types of social behavior; as the number of observed interactions grows, the set of social contacts increases; the rate at which it increases influences how we categorize their social behavior. Middle: The weight of the edges between $i$ and the other nodes represents the propensity of $i$ to interact with each of the other individuals in the group. Right: Probability distributions that correspond to the different levels of evenness in the contact propensities, both distributions are expressed by *Equation (2)*.

The online version of this article includes the following figure supplement(s) for figure 1:

**Figure supplement 1.** Outcomes using different forms of $\rho(x)$.

---

Our goal is to find a form of ρ that accurately reproduces network structure observed in real social systems. Motivated by our exploration of empirical interaction patterns from a variety of species, we propose that ρ has a power-law form:

$$\rho(x) = \frac{\phi \epsilon^\phi}{1 - \epsilon^\phi} x^{-(1+\phi)} \text{ for } \epsilon < x < 1, \tag{2}$$

where $\phi$ controls the variability in the values of $x$, and $\epsilon$ simply truncates the distribution to avoid divergence. The form of $\rho(x)$ was chosen for its analytical tractability but other heavy-tailed distributions produce a similar result (*Figure 1—figure supplement 1*). Combining (1) and (2) we find

$$\Psi(s, \phi, \epsilon) = 1 - \frac{\phi \epsilon^\phi (1-\epsilon)^{s+1}}{(1-\epsilon^\phi)(s+1)} {}_2F_1(s+1, 1+\phi, s+2, 1-\epsilon) \tag{3}$$

where the notation ${}_2F_1$ refers to the Gauss hypergeometric function (*Abramowitz and Stegun, 1975*). It follows from $\sum_j x_{j|i} = 1$ that

$$N = 1 + \frac{(1-\phi)(1-\epsilon^\phi)}{\phi \epsilon^\phi (1-\epsilon^{1-\phi})}, \tag{4}$$

which can be solved numerically to find $\epsilon$ for given values of $N$ and $\phi$. The expectation of the degree is $\kappa(s, \phi, N) = (N-1)\Psi(s, \phi, \epsilon)$.

*Figure 1* illustrates how the value of $\phi$ can produce different types of social behavior. As $\phi$ is the main determinant of social behavior in our model, we use the term *social fluidity* to refer to this quantity. Low social fluidity ($\phi \ll 1$) produces what we might describe as 'allegiant' behavior: interactions with the same partner are frequently repeated at the expense of interactions with unfamiliar individuals. As $\phi$ increases, the model produces more 'gregarious' behavior: interactions are repeated less frequently and the number of partners grows faster. While names like 'social strategy' and 'loyalty' have been applied to similar concepts (*Valdano et al., 2015*; *Miritello et al., 2013*), fluidity, as a property of matter, is a useful metaphor for communicating the main idea behind this model.

## Estimating social fluidity in empirical networks

To understand the results of the model in the context of real systems, we estimate $\phi$ in 57 networks from 20 studies of human and animal social behavior (further details in the supplement) (*Isella et al., 2011*; *Stehlé et al., 2011a*; *Mastrandrea et al., 2015*; *Vanhems et al., 2013*; *Modlmeier et al., 2019*; *Blonder and Dornhaus, 2011*; *Génois et al., 2015*; *Carter and Wilkinson, 2013*; *Grant, 1973*; *Levin et al., 2016*; *Sailer and Gaulin, 1984*; *Mourier et al., 2017*; *Massen and Sterck, 2013*; *Sade, 1972*; *Butovskaya et al., 1994*; *Takahata, 1991*; *Hass, 1991*; *Lott, 1979*; *Schein and Fohrman, 1955*; *Hobson and DeDeo, 2015*; *Gernat et al., 2018*), focusing our attention to those interactions which are capable of disease transmission (i.e. those that, at the least, require close spatial proximity). The advantage of using this model over more detailed network descriptions is that we obtain a single parameter estimate, $\phi$ that is easily compared across animal species and environments.

Each dataset provides the number of interactions that were observed between pairs of individuals. We assume that the system is closed, and that the total network size ($N$) is equal to the number of individuals observed in at least one interaction. To estimate social fluidity, we find the value of $\phi$ that minimizes $\sum_i [k_i - \kappa(s_i, \phi, N)]^2$; the total squared squared error between the observed degrees and their expectation given by the model. Uncertainty is displayed using the 2.5th and 97.5th percentile of the distribution of $\phi$ computed on a set of 1000 'bootstrap' samples, created by sampling $N$ data points, $\{k_i, s_i\}$, with replacement, from the observed data. Being estimated from the relationship between strength and degree, and not their absolute values, social fluidity is a good candidate for comparing social behavior across different systems as it is independent of the distributions of $s_i$ or $k_i$, and of the timescale of interactions.

*Figure 2* shows the estimated values of $\phi$ for all networks in our study. We organize the measurements of social fluidity by interaction type. Aggressive interactions have the highest fluidity (which implies that most interactions are rarely repeated between the same individuals), while grooming and other forms of social bonding have the lowest (which implies frequent repeated interactions between the same individuals). Social fluidity also appears to be related to species: ant systems cluster around $\phi = 1$, monkeys around $\phi = 0.5$, humans take a range of values that depend on the social environment. Sociality type does not appear to affect $\phi$; sheep, bison, and cattle have different social fluidity compared to kangaroos and bats, although they are all categorized as fission-fusion species (*Sah et al., 2018*).

Across the 57 networks, there is no evidence that social fluidity scales with the size of the network or the number of observations per individual. No correlation was found between the mean number of interactions per individual ($\bar{s}$) and social fluidity when testing for a monotonic relationship between the variables (Spearman $r^2 = 0.02$, $p = 0.36$), and in general no correlation across sets of networks taken from the same study (*Supplementary file 1: Table S2*). Similarly, network size ($N$) does not correlate with $\phi$ (Spearman $r^2 = 0.02$, $p = 0.28$). To test for a non-monotonic relationship, we partition the set of networks into 10 equally sized groups according to each of the two measures being compared, and compute the adjusted mutual information (AMI) of the two groupings. We find AMI=0.15 for the relationship between $\phi$ and $N$, and AMI=0.2 between $\phi$ and $\bar{s}$. While non-negative values of AMI typically indicate a non-random relationship, an inherent amount of clustering is to be expected in data aggregated from a diverse range of sources.

Larger values of $\phi$ correspond to higher mean degrees (Spearman $r^2 = 0.21$, $p<0.001$) and lower variability in the distribution of edge weights (measured as the index of dispersion of $w_{i,j}$; Spearman $r^2 = 0.46$, $p<0.001$). Weight variability and mean degree are uncorrelated in these data (Spearman $r^2 = 0.01$, $p = 0.54$, AMI=0.01) implying that $\phi$ combines these two entirely distinct features of social behavior. Finally, the modularity of the network (computed by the Louvain method on the unweighted network *Blondel et al., 2008*) is negatively correlated with $\phi$ ($r^2 = 0.52$, $p<0.001$). This is expected as individuals tend to be loyal to those within the same module while maintaining weaker connections with the remaining network - in all but one network the mean weight of edges within modules is higher than the mean weight of edges between modules (supplementary document).

As with any applied modeling, the validity of these results depends on the extent to which each study system conforms to the assumptions of the model. The value of $N$, for example, might not represent the true group size if some individuals in the group did not have their interactions recorded, or if there are individuals who did not interact during the time-frame of observation. While we found

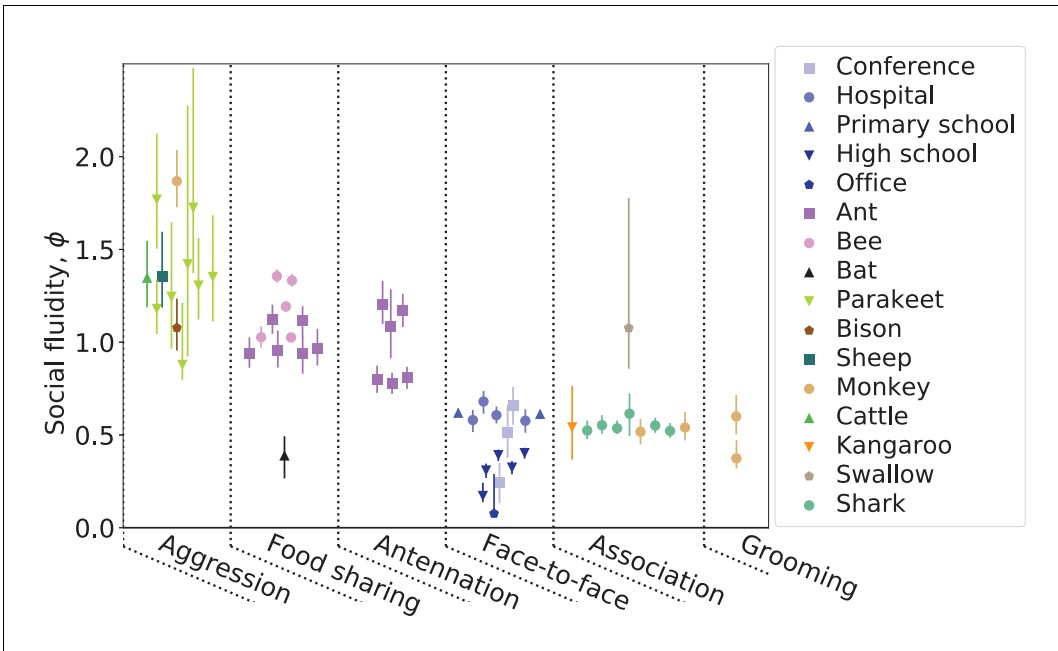

**Figure 2.** Each point represents a human or animal system for which social fluidity was estimated. Colors correspond to the species and the setting in the case of human networks. Different shapes are used as a visual aid. Lines represent the 95% bootstrap confidence interval. Results are organized by interaction type: aggression includes fighting and displays of dominance, food sharing refers to mouth-to-mouth passing of food, antennation is when the antenna of one insect touches any part of another, space sharing interactions occur with spatial proximity during foraging, face-to-face refers to close proximity interactions that require individuals to be facing each other, association is defined as co-membership of the same social group, and grooming is when one individual cleans another with their hand or other body part.

The online version of this article includes the following figure supplement(s) for figure 2:

**Figure supplement 1.** All data and fitted curves.

**Figure supplement 2.** Sensitivity to changes in time frame.

that variation in the value of $N$ did not have a large impact on the estimated value of $\phi$, as shown in *Figure 2—figure supplement 1*, we warn that the amount of consistency between model assumptions and the conditions of each study will vary, and close consideration should be given to the way data were collected when interpreting these results.

## Characterizing disease spread with social fluidity

Our objective is to characterize how social behavior influences the exposure of the group to infectious disease in a range of human and animal social systems. Intuitively, we expect an infected individual in a group with low social fluidity to expose fewer susceptible group members to the pathogen than they would in a group with highly fluid social dynamics. We explore this idea by introducing a analytical transmission model that incorporates social fluidity. Using this model, we mathematically characterize the impact of social fluidity on density dependence, and apply the model to empirical networks to predict disease spread.

## Disease transmission model

We consider the transmission of an infectious disease on the social behavior model introduced in the previous section. An infectious node $i$ interacting with a susceptible node $j$ will transmit the infection with probability β. The node will recover from infection with rate γ, assuming an exponential distribution of the length of the infectious period. The probability that the infection is transmitted from $i$ to any given $j$ is

$$T_{i \to j}(\beta, \gamma, s_i, \tau, x_{j|i}) = 1 - \exp(-s_i x_{j|i} \beta / \gamma \tau), \tag{5}$$

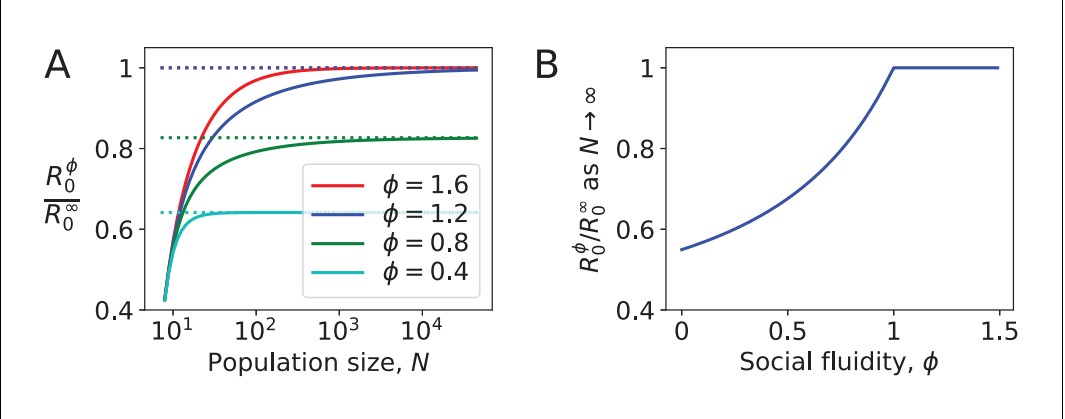

**Figure 3.** Density dependence in populations where every node has the same strength. (**A**) For different values of social fluidity, $\phi$, we show $R_0^\phi$ (from **Equation (6)**) as a function of $N$ (from **Equation (4)**) through their parametric relation with $\epsilon$. Dashed lines show the limit for large $N$. (**B**) In large populations $R_0^\phi$ increases with $\phi$ up to $\phi = 1$. Beyond this value, infections occur as frequently as they would if every new interaction occurs between a pair of individuals who have not previously interacted with each other.

assuming that the interactions $s_i$ of $i$ are distributed randomly across an observation period of duration $\tau$.

By integrating **Equation (5)** over all possible values $x_{j|i}$ and infectious period durations and multiplying by the number of susceptible individuals ($N - 1$) we obtain the expected number of infections caused by individual $i$,

$$r(s_i) = \frac{1-\phi}{\phi(\epsilon^\phi - \epsilon)}\left[1 - \epsilon^\phi + \epsilon^\phi {}_2F_1(-\phi, 1, 1-\phi; -\beta s_i/\gamma\tau) - {}_2F_1(-\phi, 1, 1-\phi; -\epsilon\beta s_i/\gamma\tau)\right]. \tag{6}$$

The basic reproductive number (usually denoted $R_0$) is defined as the mean number of secondary infections caused by a typical infectious individual in an otherwise susceptible population (**Diekmann et al., 1990**). We will use the notation $R_0^\phi$ to signify the *social fluidity reproductive number*, that is the analogue of $R_0$ derived from our social behavior model.

We assess the relation of the reproductive number with the population density by focusing on a special case where every node has the same strength, that is $s_i = s$ for all $i$, so that $R_0^\phi = r(s)$. Furthermore, we choose $\beta = \gamma\tau R_0^\infty/s$ where $R_0^\infty$ is $R_0^\phi$ as $\phi \to \infty$, that is a constant that represents what the basic reproductive number would be if every new interaction occurred between a pair of individuals who have not previously interacted with each other.

**Figure 3** shows the effect of social fluidity on the density dependence of the disease. At small population sizes, $R_0^\phi$ increases with $N$ and converges as $N$ goes to $\infty$ (**Figure 3A**). The rate of this convergence increases with $\phi$, and the limit it converges to is higher, meaning that $\phi$ determines the extent to which density affects the spread of disease. As $N \to \infty$, we find that $R_0^\phi \to R_0^\infty$ for $\phi > 1$. When $\phi < 1$, $R_0^\phi \to [(1-\phi)/\phi][{}_2F_1(-\phi, 1, 1-\phi; -R_0^\infty) - 1]$. At these values of $\phi$ the disease is constrained by individuals choosing to repeat interactions despite having the choice of infinitely many potential interaction partners (**Figure 3B**).

## Infection spread in empirical networks with heterogeneous connectivity

To apply this analogue of a reproductive number to an animal-disease system, we need to account for heterogeneous levels of social connectivity in the given population and thus the tendency for infected individuals to be those with a greater number of social partners (**Anderson et al., 1986**). For the basic reproductive number, this is often done using the mean *excess degree*, that is the degree of an individual selected with probability proportional to their degree (**Newman, 2018**). Following a similar reasoning, we define $R_0^{\mathrm{Est}}$, which incorporates the effect of social fluidity, as the

expected number of infections ($r(s_i)$) caused by an individual that has been selected with probability proportional to their degree ($k_i$):

$$R_0^{\text{Est}}(\{s_i\}, \{k_i\}, \tau, \beta, \gamma) = \frac{\sum_i k_i r(s_i)}{\sum_i k_i}. \tag{7}$$

Given the degree and strength of each individual in a network, the duration over which those interactions occurrred, and the transmission and recovery rates of the disease, we are able to estimate $\phi$, compute *Equation (6)* for each individual, and finally use *Equation (7)* to derive a statistic that provides a measure of the risk of the host population to disease outbreak.

## Numerical validation using empirical networks

We simulated the spread of disease through the interactions that occurred in the empirical data (Materials and methods). We compute $R_0^{\text{Sim}}(g)$, defined as the ratio of the number of individuals infected at the $(g+1)$-th generation to the number infected at the $g$-th generation over $10^3$ simulated outbreaks, for $g = 0, 1, 2$ ($g = 0$ refers to the initial seed of the outbreak).

*Table 1* shows the Pearson correlation coefficient and the adjusted mutual information between $R_0^{\text{Sim}}(g)$ and its corresponding value $R_0^{\text{Est}}$ obtained *Equation (7)* (Materials and methods). Equivalent results are also presented for other indicators and network statistics. The results correspond to one set of simulation conditions and are consistent across a wide range of parameter combinations (see *Supplementary file 1*). Note that a different value of β was chosen for each network to control for the varying interaction rates between networks while keeping the upper bound ($R_0^\infty$) constant (Materials and methods). While contact frequency is known to be one of the major contributors to disease risk, calibrating β in this way eliminates its effect, allowing the contribution of other network characteristics to be compared. Thus, the mean strength does not have a significant effect on $R_0^{\text{Sim}}(g)$, and higher mean edge weight does not necessarily imply higher transmission probability over the edges of the network.

These correlations support a known result regarding repeat contacts in network models of disease spread: that indicators of disease risk that are derived solely from the degree distribution are unreliable and the role of edge weights should not be neglected (*Smieszek et al., 2009*; *Stehlé et al., 2011b*). After transmission has occurred from one individual to another, repeating the same interaction serves no advantage for disease (most directly transmitted microparasites are not dose-dependent). Since a large edge weight implies a high frequency of repeated interactions, networks with a higher mean weight tend to have lower basic reproductive numbers. Furthermore, variability in the distribution of weights concentrates a yet larger proportion of interactions onto a small

**Table 1.** The Pearson correlation coefficient between quantities calculated on the network and the simulated disease outcomes (with $R_0^\infty = 3$).
Results that are significant with $p<0.01$ are labeled with *. Adjusted mutual information is calculated between the variables after partitioning the set of networks into 10 equally sized rank-order classes.

|  | Corr. with $R_0^{\text{Sim}}(g=1)$ | Adjusted MI |
| --- | --- | --- |
| $R_0^{\text{Est}}$ | 0.91* | 0.35 |
| Social fluidity | 0.73* | 0.24 |
| Excess degree | 0.64* | 0.15 |
| Mean degree | 0.53* | 0.14 |
| Network size | 0.47* | 0.18 |
| Mean strength | -0.07 | -0.02 |
| Mean clustering | -0.15 | 0.12 |
| Mean edge weight | -0.45* | 0.10 |
| Edge weight heterogeneity | -0.48* | 0.21 |
| Modularity | -0.59* | 0.12 |

number of edges, further increasing the number of repeat interactions and reducing the reproductive number.

Correlation between modularity and $R_0^{\mathrm{Sim}}(g)$ is partly due to the strong correlation between modular networks and those with high social fluidity. Consistent with other evidence (*Sah et al., 2017*), this suggests that transmission events occur mostly within the module of the seed node, with weaker social ties facilitating transmission to other modules. The effect of clustering (a measure of the number of connected triples in network *Watts and Strogatz, 1998*) correlates with smaller $R_0^{\mathrm{Sim}}(2)$, consistent with other theoretical work (*Miller, 2009*; *Smieszek et al., 2009*).

Finally, we find the model estimate of the social fluidity reproductive number $R_0^{\mathrm{Est}}$ to be, on average, within 10% of the simulated value, $R_0^{\mathrm{Sim}}(g)$ at $g = 1$. At $g = 2$ the amount of error is larger (to up to 29% for some parameter choices). Prediction accuracy at this generation is negatively correlated with the mean clustering coefficient. This is not surprising as $R_0^{\mathrm{Est}}$ does not account for the accelerated depletion of susceptible neighbors that is known to occur in clustered networks (*Miller, 2009*; *Smieszek et al., 2009*). No other properties of the network affect the accuracy of $R_0^{\mathrm{Est}}$ consistently across all parameter combinations (see *Supplementary file 1*).

## Results and discussion

We proposed a measure of fluidity in social behavior which quantifies how much mixing exists within the social relationships of a population. While social networks can be measured with a variety of metrics including size, connectivity, contact heterogeneity and frequency, our methodology reduces all such factors to a single quantity allowing comparisons across a range of human and animal social systems. Social fluidity correlates with both the density of social ties (mean degree) and the variability in the weight of those ties, although these quantities do not correlate with each other. Social fluidity is thus able to combine these two aspects seamlessly in one quantity.

By measuring social fluidity across a range of human and animal systems we are able to rank social behaviors. We identify aggressive interactions as the most socially fluid; this indicates a possible learning effect whereby each aggressive encounter is followed by a period during which individuals avoid further aggression with each other (*Parker, 1974*). At the opposite end of the scale, we find interactions that strengthen bonds (and thus require repeated interactions) such as grooming in monkeys (*Seyfarth and Cheney, 1984*) and food-sharing in bats (*Carter and Wilkinson, 2013*). The fact that food-sharing ants are far more fluid than bats, despite performing the same kind of interaction, reflects their eusocial nature and the absence of any need to consistently reinforce bonds with their kin (*Hölldobler and Wilson, 2009*).

Our results contribute to a body of work examining the disproportionate distribution of social effort in both human and animal groups. This phenomena has been directly observed in human telecommunication (*Mac Carron et al., 2016*; *Saramäki et al., 2014*; *Gonçalves et al., 2011*; *Tamarit et al., 2018*). Quantifying this aspect of sociality in animal systems, however, has been held back by the limitations of the data, such as the bias introduced by variation in activity levels across the social group (*Di Bitetti, 2000*). Additionally, while heterogeneous interaction frequencies and temporal dynamics have become common in epidemiological models (*Rocha and Blondel, 2013*; *Colman et al., 2018*), our results highlight the importance of including variability in how the individual chooses to expend their social effort.

As with most studies that aim to describe and quantify social structure, there are a number of concerns that ought to be mentioned. The degree of an individual, for example, is known to scale with the length of the observation period (*Perra et al., 2012*). This is also true of the networks used here (*Figure 2—figure supplement 1*). Similarly, social fluidity can be affected by the length of the observation window. However, since our model focuses not on the absolute value of degree, but on how degree scales with the number of observations, the results we obtain are relatively robust against this variability (*Figure 2—figure supplement 1*). Additionally, observed interactions are typically assumed to persist over time (*Perreault, 2010*). In our model this is not the case; only the distribution of edge weights remains constant, an assumption consistent with growing evidence (*Miritello et al., 2013*; *Centellegher et al., 2017*).

We therefore consider the model to be applicable to the data analysed in this study, but advise caution when applying this approach to other data sources. If the duration of a study allows for

substantial developments in the group structure, for example, then a model of edge formation and dissolution may be preferred.

Finally, we do not know the extent to which an interaction, as defined for each network, is capable of transmission which can depend on the pathogen's transmission mode and the infectious dose required. Furthermore, the transmission probability is unlikely to be the same for all interactions within the group since, for example, the duration of contact is known to be important for disease spread (*Stehlé et al., 2011b*). We did not include explicitly the duration of each contact in our model as this information was only available in a fraction of the datasets (*Barrat et al., 2014*). There is therefore potential to improve the applicability of this model as more high resolution data becomes openly available.

Our estimate of reproductive number derived from social fluidity provides a better predictor for the epidemic risk of a host population, going beyond predictors based on density or degree only. To illustrate this point, the social network of individuals at a conference ($R_0^{\text{Est}} = 1.60$; `conference_0`, supplementary document) is predicted to be at higher risk compared to the social network at a school ($R_0^{\text{Est}} = 1.39$; `highschool_0`), despite having a smaller size and lower connectivity ($N = 93$ vs. $N = 312$, and $\bar{k} = 5.63$ vs. $\bar{k} = 6.78$, respectively). The discrepancy in the risk prediction comes from the lower frequency of repeated contacts between individuals in the conference, compared to the school. Interactions between infectious individuals and those they have previously infected are redundant in terms of transmission. This dynamic is nicely captured by the social fluidity, with $\phi = 0.66$ for the conference and $\phi = 0.40$ for the high school.

Unlike previous work that explores the disease consequences of population mixing (*Volz and Meyers, 2007*; *Reluga and Shim, 2014*), our analysis allows us to investigate this relation across a range of social systems. We see, for example, how the relationship between mixing and disease risk scales with group size. For social systems that have high values of social fluidity, $R_0^{\phi}$ is highly sensitive to changes in $N$, whereas this sensitivity is not present at low values of $\phi$. This corroborates past work on the scaling of transmission being associated to heterogeneity in contact (*Begon et al., 2002*; *Ferrari et al., 2011*). Going beyond previous work, our model captures in a coherent theoretical framework both density-dependence and frequency-dependence, and social fluidity is the measure to tune from one to the other in a continuous way. Since many empirical studies support a transmission function that is somewhere between these two modeling paradigms (*Smith et al., 2009*; *Cross et al., 2013*; *Borremans et al., 2017*; *Hopkins et al., 2020*), the modeling approaches applied in this paper can be carried forward to inform transmission relationships in future disease studies.

## Materials and methods

### Python libraries

Mean clustering coefficients were computed using the *networkx* Python library. To evaluate the hypergeometric function in (3) we used the *hyp2f1* function from the *scipy.special* Python library. Numerical solutions to *Equation (4)* using the *fsolve* function from the *scipy.optimize* Python library. Adjusted mutual information was computed using *adjusted mutual info score* from the *sklearn.metrics* library. All scripts, data, and documentation used in this study are available through https://github.com/EwanColman/Social-Fluidity (*Colman, 2021*, copy archived at swh:1:rev:90b27e1b84ce4417633885cd260c89bbf1b07eac).

### Data handling

Only freely available downloadable sources of data have been used for this study. Details of the experimentation and data collection, including how the interaction type is defined, can be found through their respective publications. Here, we note some additional processes we have applied for our study.

Each human contact dataset lists the identities of the people in contact, as well as the 20 s interval of detection (*Isella et al., 2011*; *Vanhems et al., 2013*; *Stehlé et al., 2011a*; *Mastrandrea et al., 2015*; *Génois et al., 2015*). Any sequence of consecutive time intervals for which contact is detected between two individuals is considered to be one interaction. To exclude contacts detected while

participants momentarily walked past one another, only contacts detected in at least two consecutive intervals are considered interactions. Data were then separated into 24 hr subsets.

Bee trophallaxis provided experimental data for five unrelated colonies under continuous observation. We use the first hour of recorded data for each colony (*Gernat et al., 2018*). The ant trophallaxis study provided six networks: three unrelated colonies continuously observed under two different experimental conditions (*Modlmeier et al., 2019*). Ant antennation study provided six networks: three colonies, each observed for 4 hr in two sessions separated by a 2-week period. The bat study collected individual data at different times and under different experimental conditions (*Carter and Wilkinson, 2013*). For bats that were studied on more than one occasion we use only the first day they were observed.

Some data sets provided data for group membership collected through intermittent, rather than continuous, observation (*Grant, 1973*; *Massen and Sterck, 2013*; *Levin et al., 2016*; *Sailer and Gaulin, 1984*; *Mourier et al., 2017*) and typically recorded over multiple days or weeks. We construct networks from these data by recording an interaction when two individuals were seen to be in the same group during one round of observation. The shark data were divided into six datasets, each one constructed from 10 consecutive observation bouts, and spread out evenly through the 46-day period over which the data were collected.

For the grooming data (*Butovskaya et al., 1994*; *Sade, 1972*), if one animal was grooming another during one round of observations then this would be recorded as a directed interaction. Similarly for aggressive interactions (*Parker, 1974*; *Takahata, 1991*; *Hass, 1991*; *Lott, 1979*; *Schein and Fohrman, 1955*; *Hobson and DeDeo, 2015*). These data are typically collected over a period of days or weeks. When an animal was determined to be the winner of a dominance encounter then this would be recorded as a directed interaction between the winner and the loser. We consider interaction in either direction to be a contact in the network.

We considered including two rodent studies in which interaction is defined as being observed within the same territorial space (*Smith et al., 2009*; *Borremans et al., 2017*). We did not find this suitable for our analysis since the network we obtain, and the consequent results are sensitive to setting of arbitrary threshold values regarding what should, or should not, be considered sufficient contact for an interaction.

For data that did not contain the time of each interaction, contact time series were generated synthetically. For those networks, the interactions between each pair were given synthetic time-stamps in three different ways, Poisson: the time of each interaction is chosen uniformly at random from $\{0, 1, ..., 10^4\}$ seconds, Circadian: chosen uniformly at random from $\{0, 1, ..., 3333, 6666, ...., 10^4\}$, and Bursty: interaction times occur with power-law distributed inter-event times adjusted to give an expected total duration of $10^4$ seconds.

## Disease simulation

Simulations of disease spread were executed using the contacts provided by the datasets. The the bat network was omitted from this part since these data were collected over a series of independent experiments carried out at different times and under different experimental treatments.

In one run of the simulation, one seed node is randomly chosen from the network and, at a randomly selected point in time during the duration of the data, transitions to the infectious state. The duration for which they remain infectious is a random variable drawn from an exponential distribution with mean $1/\gamma$. During this time, any contact they have with other individuals who have not previously been infected will cause an infection with probability β.

The simulation runs until all individuals who were infected at the second generation of the disease, that is those infected by those infected by the seed, have recovered. The datasets are 'looped' to ensure that the timeframe of the data collection does not influence the outcome. In other words, immediately after the latest interaction, the interactions are repeated exactly as they were originally. This continues to happen until the termination criteria is met.

We set the parameters to normalise for the variation in contacts rates between networks. To achieve this, we consider a hypothetical counterpart to each network in which the strength of every node is the same, but each interaction occurs between a pair of individuals who have not previously interacted. This is equivalent to $\phi \to \infty$. Under these conditions $x_{j|i} = 1/(N-1)$ for all pairs $i, j$. It

follows that *Equation (5)* becomes $T_{i \to j} \approx s_i \beta / \gamma \tau (N-1)$, then $r(s_i) \approx s_i \beta / \gamma \tau$, and, since $k_i = s_i$ for all nodes $i$, *Equation (7)* gives

$$R_0^\infty = R_0^{\text{Est}}(\{s_i\}, \{s_i\}, \tau, \beta, \gamma) = \frac{\beta \sum_i s_i^2}{\gamma \tau \sum_i s_i} \tag{8}$$

The value of $R_0^\infty$ can be chosen arbitrarily. Then, by setting $\gamma = 1/\tau$ and $\beta = R_0^\infty \sum_i s_i / \sum_i s_i^2$ we guarantee that *Equation (8)* holds for every network. To test that our results hold over a range of disease scenarios, we repeat our analysis with $R_0^\infty = 2$, 3, and 4.

## Acknowledgements

We are grateful to Andreas Modlmeier for his involvement in the inception of this project. We are grateful for insightful feedback from Pratha Sah and Daniela Gerwehns. We also thank all the researchers who have made their behavioral data openly accessible, making this study possible.

## Additional information

### Funding

| Funder | Grant reference number | Author |
| --- | --- | --- |
| National Science Foundation | Award 141429 | Ewan Colman<br>Ephraim M Hanks<br>David P Hughes<br>Shweta Bansal |

The funders had no role in study design, data collection and interpretation, or the decision to submit the work for publication.

### Author contributions

Ewan Colman, Conceptualization, Data curation, Software, Formal analysis, Investigation, Visualization, Methodology, Writing - original draft, Writing - review and editing; Vittoria Colizza, Methodology, Writing - review and editing; Ephraim M Hanks, Funding acquisition, Methodology, Writing - review and editing; David P Hughes, Funding acquisition, Writing - review and editing; Shweta Bansal, Conceptualization, Supervision, Funding acquisition, Investigation, Methodology, Writing - review and editing

### Author ORCIDs

Ewan Colman https://orcid.org/0000-0003-2551-8589
Vittoria Colizza http://orcid.org/0000-0002-2113-2374
David P Hughes https://orcid.org/0000-0002-9954-8919
Shweta Bansal https://orcid.org/0000-0002-1740-5421

### Decision letter and Author response

Decision letter https://doi.org/10.7554/eLife.62177.sa1
Author response https://doi.org/10.7554/eLife.62177.sa2

## Additional files

### Supplementary files

- Supplementary file 1. Supplementary tables.
- Supplementary file 2. Results for all networks.
- Transparent reporting form

## Data availability

All scripts, data, and documentation used in this study are available through https://github.com/EwanColman/Social-Fluidity (copy archived at https://archive.softwareheritage.org/swh:1:rev:90b27e1b84ce4417633885cd260c89bbf1b07eac).

The following previously published datasets were used:

| Author(s) | Year | Dataset title | Dataset URL | Database and Identifier |
|---|---|---|---|---|
| Modlmeier AP, Colman E, Hanks EM, Bringenberg R, Bansal S, Hughes DP | 2019 | Ant colonies maintain social homeostasis in the face of decreased density | https://doi.org/10.5061/dryad.sh4m4s6 | Dryad Digital Repository, 10.5061/dryad.sh4m4s6 |
| Hobson EA, DeDeo S | 2016 | Social feedback and the emergence of rank in animal society | https://doi.org/10.5061/dryad.p56q7 | Dryad Digital Repository, 10.5061/dryad.p56q7 |
| Mourier J, Brown C, Planes S | 2016 | Learning and robustness to catch-and-release fishing in a shark social network | https://doi.org/10.5061/dryad.gg859 | Dryad Digital Repository, 10.5061/dryad.gg859 |
| Carter CG, Wilkinson GS | 2013 | Food sharing in vampire bats: reciprocal help predicts donations more than relatedness or harassment | https://doi.org/10.5061/dryad.tg7b1 | Dryad Digital Repository, 10.5061/dryad.tg7b1 |
| Levin II, Zonana DM, Fosdick BK, Song SJ, Knight R, Safran RJ | 2016 | Stress response, gut microbial diversity and sexual signals correlate with social interactions | https://doi.org/10.5061/dryad.3jn35 | Dryad Digital Repository, 10.5061/dryad.3jn35 |
| Gelardi V, Godard J, Paleressompoulle D, Claidière N, Barrat A | 2008 | Baboons' interactions | http://www.sociopatterns.org/datasets/baboons-interactions/ | Sociapatterns, baboons-interactions |

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
