## [Decision Letter]

**Acceptance summary:**

This manuscript makes an important point for studies of contagion in both human and animal populations. This paper provides a way of characterising heterogeneity in social systems. Furthermore, the proposed measure of social fluidity can be used to distinguish between different types of animal social systems. Hence, the measure is of relevance for studies of human and animal social networks.

**Decision letter after peer review:**

Thank you for submitting your article "Social fluidity mobilizes contagion in human and animal populations" for consideration by *eLife*. Your article has been reviewed by 3 peer reviewers, including Niel Hens as the Reviewing Editor and Reviewer #3, and the evaluation has been overseen by Miles Davenport as the Senior Editor. The following individual involved in review of your submission has agreed to reveal their identity: Jari Saramaki (Reviewer #2).

The reviewers have discussed the reviews with one another and the Reviewing Editor has drafted this decision to help you prepare a revised submission.

Please pay particular attention to the comments by Reviewer #1: items 1 and 2 and Reviewer #3: item 1 which focus on the relevance of this work. The other comments address punctual issues and/or clarifications; they should be looked at carefully and it would be good to organise your reply based on topics rather than a point-by-point reply.

*Reviewer #1:*

This is a well written paper on an interesting and important topic, characterizing the social contact frequency and network degree into a single measure, social fluidity. The concept is explained and mathematically derived, then applied to an analysis of 50 network data sets spanning 13 human and animal species. The authors then apply that parameter into a mathematical model for infectious disease dynamics to characterize its impact on transmission potential, via R. This is useful in developing understanding of the measure, and demonstrating its applicability in epidemiology. I have only relatively minor suggestions, mostly related to clarifying ideas and terms in the manuscript.

1. The main concern is the treatment of time within the data and fluidity measure. The authors appropriate note this as a limitation in the Discussion section. Because the inputs to fluidity depend on degree and within-node contacts, the measurement time frame for both in each dataset is critical. The authors cite a paper to support the idea that degree scales with the observation length (citation 59), but that is a limited analysis of 3 human networks (2 of which were online convenience samples) that may not apply more broadly. It would be helpful to have more clarification on the range of time frames for data collection across the datasets to understand whether weighted cross-sectional networks are the best modeling approach here (versus modeling dynamic networks of edge formation and dissolution). This may be important in the comparative analysis of different contact types (aggression contacts versus grooming contacts).

2. The authors establish the importance of the fluidity measure for comparative empirical research well, but It would also be helpful to have further clarity on its advantages for modeling over current approaches that might model network structure with two or more parameters. What are the broader benefits of this approach?

3. It was not clear how the contact types (aggression versus grooming, eg) were defined. Were these a component of the secondary data, or did the current authors use a classification scheme? Some further details on the measurement of the data would be helpful.

4. What are the implications of treating the system as closed and the network consisting only of non-isolates on the empirical comparisons and the epidemic modeling? These are assumptions required by the data, but not always realistic and could have a meaningful impact on the outcomes (e.g., non-differential misclassification for aggression contacts that may change the network structure, or include many isolates). The importance of these assumptions may depend on the measurement timeframe (i.e., less important if short observations).

*Reviewer #2:*

This manuscript makes an important point for studies of contagion in both human and animal populations: the heterogeneity of contact frequencies matters a lot. The individual-level heterogeneity of weights/contact frequencies in egocentric networks is nicely captured by the concept of social fluidity and the model parametrised by $\phi$ whose fitted values clearly differ for datasets from different species (Figure 2). Finally, the spreading model shows that $\phi$ has clear effects on R0 – the effect of ego-network weight heterogeneity on disease transmission is something that I have hypothesised myself as well, so I congratulate the authors for getting there first!

In my view, this paper makes several important contributions: in addition to the context of contagious disease, it provides a way of characterising heterogeneity in social systems and shows that it even works for distinguishing human contact networks under different circumstances (the Sociopatterns data sets). Furthermore, the proposed measure of social fluidity can be used to distinguish between different types of animal social systems. Hence, the measure is of relevance for studies of human and animal social networks.

As the science is solid, the results are important, and the manuscript is well and clearly written, I recommend publishing it, after some fairly straightforward clarifications/modifications.

1) Would it be possible to justify the choice of the power-law form for Equation (2)? And would the results be sensitive to this choice – would using something like a log-normal or stretched exp yield similar results? (Intuitively, my expectation is that the exact form of the distribution should not matter too much as N is fairly low in all studied cases, so whatever is mathematically the most convenient distribution should be fine).

2) page 3, bottom left column: "There is no significant correlation between the mean number of interactions per individual (s) and social fluidity…" Has r^2 been calculated over all datasets or separately for one species over their respective data sets?

"…which implies that sampling bias does not affect the estimation of social fluidity". How this is to be interpreted depends on the answer to the above, but I am not certain if one can make this statement, at least if the correlation is over all species/datasets. I would think that it is difficult to escape some sampling bias (as for most network measures…), unless one has several samples of different size for the same species under the same circumstances, and can show in those samples that $\phi$ doesn't depend on N.

"Similarly, network size does not correlate with $\phi$…" Again, is the correlation over all species?

3) In the subsection "Numerical validation using empirical networks" it is stated that "Since a large edge weight implies a high frequency of repeated interactions, networks with a higher mean weight tend to have lower basic reproductive number. Furthermore, variability in the distribution of weights…"

Would the variability not be a requirement for a higher mean weight leading to lower R0, so that the cause of the lower R0 is the combination of higher weights and high variability? If one considers two networks with uniform weights that are otherwise identical but one has twice the mean weight, would that one not have a *higher* R0?

4) Discussion: "We see, for example, how the relationship between mixing and disease risk scales with population density. For social systems that have high values of social fluidity, $R_0^\phi$ is highly sensitive to changes in N…" Is N conceptually the same as population density? Would, under the network paradigm, the average degree be a better proxy of population density?

Reviewer #3:

The authors define the concept of social fluidity to better define how social behaviour influences contagion process in human and animal populations. Whereas I believe the manuscript is well written, its current version requires a few clarifications.

1. I think it's important to mention that the concept of social fluidity hasn't been tested in relation to infectious disease data. Does it provide a good/better fit to infectious disease data as compared to assumptions of frequency and density dependent mass action etc.

2. In the social behavior model: the authors use frequency for the edge weight; should weighing not be done on the basis of risk assessment of these interactions?

3. Please better motivate the use of the power-law form in equation (2). Have the authors considered alternatives?

[Editors' note: further revisions were suggested prior to acceptance, as described below.]

Thank you for resubmitting your work entitled "Social fluidity mobilizes contagion in human and animal populations" for further consideration by *eLife*. Your revised article has been evaluated by Miles Davenport (Senior Editor) and a Reviewing Editor.

The manuscript has been improved but there are some remaining issues that need to be addressed, as outlined below:

– In terms of the authors' reply on 'Was this relationship linear … ': Given that a Pearson correlation coefficient assumes linearity holds, one cannot measure strength of association in case of non-linearity. Please verify or use an alternative measure (see e.g. https://www.pnas.org/content/111/9/3354).

– The authors use 100 bootstraps to quantify the uncertainty of phi; this seems small to me; why not use 1000 bootstraps (as well as assessing whether or not estimates of 5% and 95% percentiles are stable for that number)?

---

## [Author Response]

Please pay particular attention to the comments by Reviewer #1: items 1 and 2 and Reviewer #3: item 1 which focus on the relevance of this work. The other comments address punctual issues and/or clarifications; they should be looked at carefully and it would be good to organise your reply based on topics rather than a point-by-point reply.

As suggested, we start by detailing our response and modifications to the manuscript that were mentioned by more than one reviewer, and then address our remaining responses to each individual reviewer.

Point 1: Power law

Reviewer 2: "1) Would it be possible to justify the choice of the power-law form for Equation (2)? And would the results be sensitive to this choice – would using something like a log-normal or stretched exp yield similar results? (Intuitively, my expectation is that the exact form of the distribution should not matter too much as N is fairly low in all studied cases, so whatever is mathematically the most convenient distribution should be fine)"Reviewer 3: "3. Please better motivate the use of the power-law form in equation (2). Have the authors considered alternatives?"

It is true that the exact form of this distribution is not important, and other heavy-tailed distributions will give similar results. We show this is in a supplementary analysis (Figure 1-supplement 1) by simulating the interactions of one individual whose interactions with other members of the group are determined by probabilities drawn from other distributions. Figure 1-supplement 1 shows that results obtained from log-Normal and power-law distributions look very similar and can cover a similar range of behaviours.

In the main text this is referenced in the "Social behavior model" section. When introducing the power-law it now says:

“The form of $ρ(x)$ was chosen for its analytical tractability but other heavy-tailed distributions produce a similar result (Figure S2).”

Point 2: Scope of this work

Reviewer 1: "The authors establish the importance of the fluidity measure for comparative empirical research well, but It would also be helpful to have further clarity on its advantages for modeling over current approaches that might model network structure with two or more parameters. What are the broader benefits of this approach?"Reviewer 3: "1. I think it's important to mention that the concept of social fluidity hasn't been tested in relation to infectious disease data. Does it provide a good/better fit to infectious disease data as compared to assumptions of frequency and density dependent mass action etc."

We feel that the strength of our work resides in (i) introducing a concept, named social fluidity and expressed in mathematical form, that captures the distribution of interactions of individual hosts with varying frequency and strength, and (ii) showing that social fluidity is a stronger predictor of epidemic outcomes than commonly used metrics. While this specific study was not meant to fit the social fluidity model to infectious disease data, social fluidity was estimated from empirical networks and its role in disease spread was compared to simulated disease data on the original empirical networks. Finally, the theory we introduce is able to seamlessly connect between density-dependent and frequency-dependent approaches, so far considered independent and resting on disjoint frameworks.

Both comments suggest that we need to clarify the scope and purpose of this work. Firstly, we have added additional sentences to the abstract to emphasize the novelty…

“Large values of social fluidity correspond to gregarious behavior whereas small values signify the existence of persistent bonds between individuals.”

And the contribution to the field…

“…we demonstrate that social fluidity is a stronger predictor of disease outcomes than both group size and connectivity, and it provides an integrated framework for both density-dependent and frequency-dependent transmission.”

We have also changed the third paragraph of the introduction to better motivate the purpose of our study and the current gap in the literature that it addresses…

“While transmission rate has been observed to scale non-linearly with group size for a number of disease systems, it remains unclear how much this dependency is related to the internal social structure of the group; few studies observe social dynamics at sufficient detail while simultaneously monitoring the disease status of each individual. In the absence of direct observations, our contribution to this discussion centers around modelling; incorporating empirical social data with computational simulations. We address two specific questions. Firstly, can we quantify the variability in how individuals choose to distribute their social effort within a group, and secondly, what will this tell us about the effect that population density has on disease transmission?”

Additionally, in the first paragraph of "Estimating social fluidity in empirical networks" we mention one specific advantage of the model over models that have more parameters…

“The advantage of using this model over more detailed network descriptions is that we obtain a single parameter estimate, $\phi$ that is easily compared across animal species and environments.”

Reviewer #1:1. The main concern is the treatment of time within the data and fluidity measure. The authors appropriate note this as a limitation in the Discussion section.

We thank the reviewers for this comment. We have addressed this with a supplementary analysis of the networks for which temporal information are available (Figure 2-supplement 2). Briefly, it is a sensitivity analysis to see the effects of using a shorter time frame. References to this analysis are added to various parts of the main text and correspond to your more specific remarks as follows….

Because the inputs to fluidity depend on degree and within-node contacts, the measurement time frame for both in each dataset is critical.

Our supplementary analysis presents the values of phi calculated when only the first 50% of the observations are used. The estimation of social fluidity is largely insensitive to the sampling time frame, using two different methods of estimation. This is referenced in the "Discussion" section …

“However, since our model focuses not on the absolute value of degree, but on how degree scales with the number of observations, the results we obtain are relatively robust against this variability (Figure S3A)”

The authors cite a paper to support the idea that degree scales with the observation length (citation 59), but that is a limited analysis of 3 human networks (2 of which were online convenience samples) that may not apply more broadly.

We checked this for the networks used in our study and include the results in the same supplementary figure. At the same point in the discussion we now add …

“This is also true of the networks used here (Figure 2 supplement 2).”

It would be helpful to have more clarification on the range of time frames for data collection across the datasets to understand whether weighted cross-sectional networks are the best modeling approach here (versus modeling dynamic networks of edge formation and dissolution). This may be important in the comparative analysis of different contact types (aggression contacts versus grooming contacts).

The time frames for data collection do vary greatly and this is something we want to be clear about. We include details of the datasets in the "Data handling" part of the "Materials and methods" section. In the "Discussion" section we mention this concern…

“We therefore consider the model to be applicable to the data analysed in this study, but advise caution when applying this approach to other data sources. If the duration of a study allows for substantial developments in the group structure, for example, then a model of edge formation and dissolution may be preferred.”

3. It was not clear how the contact types (e.g. aggression versus grooming) were defined. Were these a component of the secondary data, or did the current authors use a classification scheme? Some further details on the measurement of the data would be helpful.

These are all defined by the original studies. We have added a sentence to the "Data handling" part of the "Materials and methods" section to make this clear.

4. What are the implications of treating the system as closed and the network consisting only of non-isolates on the empirical comparisons and the epidemic modeling? These are assumptions required by the data, but not always realistic and could have a meaningful impact on the outcomes (e.g., non-differential misclassification for aggression contacts that may change the network structure, or include many isolates). The importance of these assumptions may depend on the measurement timeframe (i.e., less important if short observations).

We agree with the reviewer that isolates (individuals that do not interact during the time frame of observation) are an important data limitation to consider as they can affect the value of N (group size). In our supplementary analysis we tested this directly by varying the time frame of observation, and thus changing the number of isolates.

The following added to end of the section "Estimating social fluidity in empirical networks"

“As with any applied modelling, the validity of these results depends on the extent to which each study system conforms to the assumptions of the model. The value of $N$, for example, might not represent the true group size if some individuals in the group did not have their interactions recorded, or if there are individuals who did not interact during the time-frame of observation. While we found that variation in the value of $N$ did not have a large impact on the estimated value of $\phi$, as shown in Figure S3D, we warn that the amount of consistency between model assumptions and the conditions of each study will vary, and close consideration should be given to the way data were collected when interpreting these results.”

Reviewer #2:2) page 3, bottom left column: "There is no significant correlation between the mean number of interactions per individual (s) and social fluidity…" Has r^2 been calculated over all datasets or separately for one species over their respective data sets?

The results were presented for all species, and we have added a set of supplementary tables containing correlations between all the social variables within species. This is referenced in the section "Estimating social fluidity in empirical networks" as follows …

Across the $57$ networks, there is no significant correlation between the mean number of interactions per individual ($\bar{s}$) and social fluidity (Pearson $r^{2}=0.02$, $p=0.27$), and in general no correlation across sets of networks taken from the same study (Tables S2)."…which implies that sampling bias does not affect the estimation of social fluidity". How this is to be interpreted depends on the answer to the above, but I am not certain if one can make this statement, at least if the correlation is over all species/datasets. I would think that it is difficult to escape some sampling bias (as for most network measures…), unless one has several samples of different size for the same species under the same circumstances, and can show in those samples that $\phi$ doesn't depend on N.

We thank the reviewer for this comment – we have removed that statement now. Instead we say:

“Thus there is no evidence that the number of observations per individual affects the estimated value of $\phi$.”

"Similarly, network size does not correlate with $\phi$…" Again, is the correlation over all species?

As before, we have now added that information in the supplement. The main results still hold.

3) In the subsection "Numerical validation using empirical networks" it is stated that "Since a large edge weight implies a high frequency of repeated interactions, networks with a higher mean weight tend to have lower basic reproductive number. Furthermore, variability in the distribution of weights…"Would the variability not be a requirement for a higher mean weight leading to lower R0, so that the cause of the lower R0 is the combination of higher weights and high variability? If one considers two networks with uniform weights that are otherwise identical, but one has twice the mean weight, would that one not have a ‘higher’ R0?

An important part of the disease model is the calibration of the β parameter to control for the effect of contact frequency, which varies greatly between different networks. Since mean weight is closely related to the contact frequency (i.e. the mean strength divided by the length of the time frame) choosing β in the way we do causes networks with high mean weight to have lower values of β (all else being equal). In the example you provide, the network with higher mean weight would indeed have lower R0, since the mean strength would also double and the β we apply would be halved.

We have explained this in a bit more depth in the revised version:

“While contact frequency is known to be one of the major contributors to disease risk, calibrating $\β$ in this way eliminates its effect, allowing the contribution of other network characteristics to be compared. Thus, the mean strength does not have a significant effect on $R_{0}^{\text{Sim}}(g)$, and higher mean edge weight does not necessarily imply higher transmission probability over the edges of the network.”

4) Discussion: "We see, for example, how the relationship between mixing and disease risk scales with population density. For social systems that have high values of social fluidity, $R_0^\phi$ is highly sensitive to changes in N…" Is N conceptually the same as population density? Would, under the network paradigm, the average degree be a better proxy of population density?

We have changed "population density" to "group size" to more accurately reflect what we found.

1) Figure 2 – it took me a while to understand that there are several data sets for each species (or Sociopatterns setting), so this could be mentioned in the caption.

2) This is probably an error in production, but there is a full-page figure (k vs s for the different datasets) at the end of the PDF without any caption that is also not referred to in the text (unless I missed it).

Thank you for pointing these things out. We have added "Colours correspond to the species and the setting in the case of human networks" to the figure caption. The supplementary figure 2 is now included in the new supplementary document.

Reviewer #3:2. In the social behavior model: the authors use frequency for the edge weight; should weighing not be done on the basis of risk assessment of these interactions?

In general, yes, we agree that it makes sense to define "interaction" in terms of a disease. In our study, however, we do not specify any particular disease. Specifically, in the section "Estimating infection spread in empirical networks with heterogeneous connectivity" we specify how the transmission risk parameter (β) is selected:

“… value of $\β$ was chosen for each network to control for the varying interaction rates between networks … While contact frequency is known to be one of the major contributors to disease risk, calibrating $\β$ in this way eliminates its effect, allowing the contribution of other network characteristics to be compared.”

To address your concern we have added the following to the part of "Discussion" that mentions the limitations of this study:

“Finally, we do not know the extent to which an interaction, as defined for each network, is capable of transmission which can depend on the pathogen's transmission mode and the infectious dose required.”

[Editors' note: further revisions were suggested prior to acceptance, as described below.]

The manuscript has been improved but there are some remaining issues that need to be addressed, as outlined below:– In terms of the authors' reply on 'Was this relationship linear … ': Given that a Pearson correlation coefficient assumes linearity holds, one cannot measure strength of association in case of non-linearity. Please verify or use an alternative measure (see e.g. https://www.pnas.org/content/111/9/3354).

We want to know if the size of the network, or the number of observations in the sample, has an effect on our introduced measure, social fluidity. We therefore think it is appropriate to test for a monotonic relationship, and have switched from using the Pearson coefficient to the Spearman coefficient, which yields similar results. We further address whether there may be a non-monotonic relationship using mutual information as suggested.

This is now addressed in the section "Estimating social fluidity in empirical networks" as follows:

Across the $57$ networks, there is no evidence that social fluidity scales with the size of the network or the number of observations per individual. No correlation was found between the mean number of interactions per individual ($\bar{s}$) and social fluidity when testing for a monotonic relationship between the variables (Spearman $r^{2}=0.02$, $p=0.36$), and in general no correlation across sets of networks taken from the same study (Tables S2). Similarly, network size ($N$) does not correlate with $\phi$ (Spearman $r^{2}=0.02$, $p=0.28$). To test for a non-monotonic relationship, we partition the set of networks into $10$ equally sized groups according to each of the two measures being compared, and compute the adjusted mutual information (AMI) of the two groupings. We find AMI=$0.15$ for the relationship between $\phi$ and $N$, and AMI=$0.2$ between $\phi$ and $\bar{s}$. While non-negative values of AMI typically indicate a non-random relationship, an inherent amount of clustering is to be expected in data aggregated from a diverse range of sources.

We have also added the measure of mutual information to Table 1 for comparing analytical and simulated values of R0. Note that we still use the Pearson coefficient here as we want to show that this relationship is linear.

– The authors use 100 bootstraps to quantify the uncertainty of phi; this seems small to me; why not use 1000 bootstraps (as well as assessing whether or not estimates of 5% and 95% percentiles are stable for that number)?

We have increased the number of bootstrap samples to 1000 and updated Figure 2.